# Histone variant H2A.Z regulates zygotic genome activation

Dafne Ibarra-Morales [1,2], Michael Rauer [1], Piergiuseppe Quarato [3], Leily Rabbani[1], Fides Zenk[1,2,4], Mariana Schulte-Sasse[1,2], Francesco Cardamone [1,2], Alejandro Gomez-Auli [1], Germano Cecere [3] & Nicola Iovino [1]✉

During embryogenesis, the genome shifts from transcriptionally quiescent to extensively active in a process known as Zygotic Genome Activation (ZGA). In *Drosophila*, the pioneer factor Zelda is known to be essential for the progression of development; still, it regulates the activation of only a small subset of genes at ZGA. However, thousands of genes do not require Zelda, suggesting that other mechanisms exist. By conducting GRO-seq, HiC and ChIP-seq in *Drosophila* embryos, we demonstrate that up to 65% of zygotically activated genes are enriched for the histone variant H2A.Z. H2A.Z enrichment precedes ZGA and RNA Polymerase II loading onto chromatin. In vivo knockdown of maternally contributed Domino, a histone chaperone and ATPase, reduces H2A.Z deposition at transcription start sites, causes global downregulation of housekeeping genes at ZGA, and compromises the establishment of the 3D chromatin structure. We infer that H2A.Z is essential for the *de novo* establishment of transcriptional programs during ZGA via chromatin reorganization.

[1] Max Planck Institute of Immunobiology and Epigenetics, 79108 Freiburg im Breisgau, Germany. [2] University of Freiburg, Faculty of Biology, 79108 Freiburg im Breisgau, Germany. [3] Institut Pasteur, Mechanisms of Epigenetic Inheritance, Department of Developmental and Stem Cell Biology, UMR3738, CNRS, 75724 Paris, Cedex 15, France. [4] Present address: Department of Biosystems Science and Engineering ETH (D-BSSE ETH Zürich), 4058 Basel, Switzerland. ✉email: iovino@ie-freiburg.mpg.de

At fertilization, all cellular processes are governed by maternally provided mRNAs and proteins. Transcription of the zygotic genome commences hours to days after fertilization, depending on the organism, in a process known as Zygotic Genome Activation (ZGA)[1,2]. At ZGA, developmental control is transferred from maternally provided to zygotically encoded factors. ZGA requires the coordinated expression of thousands of genes in a timely manner and is essential for embryo survival. In *Drosophila*, zygotic transcription initiates at embryonic nuclear cycle 8 (NC8) with the activation of about 100 genes[3–5]. At NC14, nearly a third of all *Drosophila* genes (corresponding to around 6000 genes) start to be transcribed, an event known as major wave of ZGA.

The de novo establishment of transcriptional programs is thought to be driven primarily by pioneer factors[1,6,7]. These transcription factors can bind their target sites within closed chromatin, thereby rendering it accessible to other factors[8]. In *Drosophila*, the pioneer transcription factor Zelda is required to activate hundreds of genes before and during the major wave of ZGA[7,9,10]. However, these genes represent only a fraction of the thousands of genes that become transcriptionally activated at this stage, suggesting that other factors or mechanisms activate the majority of genes throughout ZGA.

Changes in chromatin state are known to influence gene expression programs during diverse transitions, such as oocyte specification, embryo patterning and germ layer specification[11–14]. Global alterations in core histone levels for example have been shown to affect ZGA timing and transcription in several organisms such as flies, zebrafish and frogs[15–17]. Substantial evidence in both vertebrate and invertebrate models shows that major chromatin remodeling occurs immediately before ZGA[18,19]. This chromatin reorganization involves histone posttranslational modifications, the incorporation of histone variants, and changes in the 3D chromatin structure of the genome[19–24]. In *Drosophila*, maternally inherited H3K27me3 and H4K16ac are important for the correct repression or activation, respectively, of their target genes at ZGA[20,22]. Moreover, a mutation of the embryonic-specific H1 histone variant (BigH1) causes premature ZGA onset in flies and its role becomes particularly relevant upon challenging conditions such as low temperature[21,25]. Along these lines, the histone variant H3.3 is also known to be essential for early development, since its incorporation in the paternal genome just after fertilization is required for the progression of development[26]. However, to date it remains unclear to what extent chromatin state and structure regulates ZGA.

The histone variant H2A.Z is one of the most ancient histones, present from yeast to mammals[27]. Histone H2A.Z represents around 4~10% of the total H2A pool[28] and has been implicated in multiple processes such as transcriptional control, DNA repair, heterochromatin formation and genetic stability[29–32]. H2A.Z is incorporated into the chromatin throughout the genome and it is known to regulate transcription in a context-dependent manner[29,32–35]. Nevertheless, H2A.Z studies during early development have been hindered by the lethality associated with its mutation in organisms such as frogs[36], flies[37] and mouse[38].

Some insight about the developmental role of H2A.Z comes from studies in flies, where the maternal protein Jabba anchors the canonical histones H2A, H2B and the histone variant H2A.Z to lipid droplets[39], mechanism that is important for H2A.Z buffering in pre-ZGA stages[40]. Mutant embryos for Jabba are sensitive to changes on histone levels, such as *SLBP* mutation or extra doses of H2A.Z. They also possess an elevated H2A.Z/H2A ratio in the nucleus, increased DNA damage and reduced viability[39,40]. Nonetheless, it is still undefined if there is a direct effect of H2A.Z on ZGA.

Here, we used a combination of genomic and genetic approaches to investigate the function of histone variants and histone modifications during ZGA in *Drosophila*, focusing on the histone variant His2Av (herein after referred to as H2A.Z). ChIP-seq revealed that most genes that become activated at ZGA, which are functionally independent of the pioneer factor Zelda, are enriched for H2A.Z. Importantly, the deposition of this histone variant precedes ZGA. We show that germline-specific knockdown of maternally provided Domino (DomKD), a known H2A.Z histone chaperone and ATPase, is embryonic lethal. DomKD embryos lack H2A.Z at the transcription start sites and display global downregulation of housekeeping genes during ZGA. Interestingly, we find that Domino is required to maintain the insulation of topologically associated domains but not nucleosome positioning. Our data suggest that H2A.Z is necessary for gene activation at ZGA and therefore for correct zygotic transcriptional programming.

## Results

### The majority of active transcription start sites (TSS) at zygotic genome activation are enriched for H2A.Z and do not depend on Zelda

To investigate the role of chromatin organization in ZGA regulation, we manually selected ZGA embryos and conducted Global Run-On sequencing (GRO-seq). This approach allowed us to detect zygotic transcription and exclude maternal transcripts. To avoid redundancy, we selected unique promoters (transcription units with non-overlapping promoters, see Methods for details) for further analysis. In total, we obtained 7227 zygotic promoters, corresponding to 6249 genes (Fig. 1a and Supplementary Fig. 1a). As expected, these genes exhibited strong binding by RNA polymerase II (Pol II) core subunit Rpb3 and open chromatin at the TSS, when compared to transcriptionally inactive genes (Supplementary Fig. 1a).

We divided this set of zygotic promoters into two groups: (i) promoters that are dependent on the pioneer factor Zelda (Zld) for Pol II occupancy (Zld-dependent promoters) and (ii) those that are not (Zld-independent promoters) (according to ref. [10]). Of the 7,227 active promoters, 713 (569 genes) were Zld-dependent, whereas 6514 promoters (5680 genes) were Zld-independent (Fig. 1a and Supplementary Fig. 1a). To compare the chromatin states associated with Zld-independent genes and Zld-dependent genes, we used a broad panel of genomic approaches, such as ChIP-seq, ATAC-seq and HiC (Supplementary Fig. 1a, c). Interestingly, both groups of genes shared many features, such as displaying similar patterns of chromatin accessibility and of the histone marks H3K4me3, H3K27ac, H3K36me3. However, Zld-independent and Zld-dependent genes showed significant differences in the enrichment of the histone variant H2A.Z on the promoter region and in nucleosome positioning (particularly, in the +1 nucleosome) (Fig. 1a and Supplementary Fig. 1a). Indeed, H2A.Z was nearly absent from Zld-dependent genes but highly enriched in Zld-independent genes. Of the 6514 Zld-independent promoters, 4541 were positive for H2A.Z (corresponding to 4053 genes). Moreover, Zld-independent genes containing H2A.Z had well-positioned nucleosomes (Fig. 1a), in contrast to Zld-dependent genes. Thus, we further divided our set of Zld-independent zygotic genes into two different groups: 1) H2A.Z-positive genes and 2) H2A.Z-negative genes (See Methods and Supplementary Data 1). We found no evidence of H2A.Z enrichment at enhancer regions using the Vienna Tiles collection[41] (Supplementary Fig. 1b). We concluded that H2A.Z localizes mainly in promoters at this developmental stage, and focused on this for further analysis.

3D chromatin structure is also established for the first time in the developing embryo at ZGA when topologically associated

domains (TADs) and A- and B-compartments form[23,24,42]. TADs are submegabase - to - megabase long regions that show a high frequency of interactions[43,44]. Interaction frequencies can be inferred from the insulation score, which represents the average of interactions frequencies crossing over a defined genomic region[43]. We aimed to determine whether Zld-dependent genes, H2A.Z-positive genes and H2A.Z-negative genes have different 3D chromatin structures, in particular, different TAD structures. To test this, we measured insulation scores over the TSSs and found that Zld-dependent loci had the highest insulation and H2A.Z-negative loci had the lowest (Supplementary Fig. 1c). In congruence with this, Zld-dependent TSSs were the closest to a TAD boundary, whereas H2A.Z negative genes were more distant (Supplementary Fig. 1d). Taken together, these results indicate that the genes activated at ZGA have structurally different chromatin and spatial environments, defined by Zld and the presence of H2A.Z.

To determine if H2A.Z is deposited at genes prior to zygotic transcription initiation, we examined pre-ZGA embryos (NC8-12). Immunofluorescence experiments revealed that H2A.Z deposition occurs before ZGA (Fig. 1b, left). H2A.Z localized in the nucleus as early as NC9 to 12, albeit with a weak signal, and remained in the nucleus throughout ZGA (Fig. 1b, right and Supplementary Fig. 1e). Before ZGA, H2A.Z could also be seen in small ring structures in the cytoplasm, corresponding to lipid droplets; in these early stages, we found H2A.Z signal also on mitotic chromosomes (Supplementary Fig. 1e, f), as reported by[39,40]. As expected, pre-ZGA embryos displayed Pol II binding only at the ~100 early genes which are known to be transcribed between NC8 to 10, as shown by ChIP-seq (Fig. 1c and Supplementary Fig. 1g, h). Similar to H2A.Z deposition, nucleosomes were already well-positioned in pre-ZGA embryos (Fig. 1c). Thus, H2A.Z is not only enriched on actively transcribed genes, but also deposited prior to Pol II complex binding.

**Domino is the main histone chaperone for H2A.Z on TSS during ZGA.** To unbiasedly identify maternally supplied histone chaperones that deposit H2A.Z on the TSS, we performed immunoprecipitation-mass spectrometry (IP-MS) using *Drosophila* ovaries expressing a H2A.Z-FlagHA transgene. The H2A.Z-FlagHA transgene is expressed exclusively in the germline and not in the accessory somatic tissues of the germline (see Methods) (Fig. 2a and Supplementary Data 2). We uncovered the chromatin remodeler and histone chaperone Domino (isoforms A and B) as a strong candidate H2A.Z chaperone (Fig. 2b, magenta dots), consistent with work showing that Domino and its orthologs bind specifically to the H2A.Z variant in *Drosophila*[34,45], yeast[46] and mammals[47]. Domino is a SWR1-type ATPase, homolog of the human proteins SRCAP and p400, and its ATPase activity is known to be required for H2A.Z deposition[34,45,48]. Consistent with previous studies showing that *Drosophila* Domino is required within the Tip60 complex for H2A.Z turnover during DNA double-strand break repair[45], we found several members of this complex (YL-1, DMAP-1, Brd8, Bap55, Gas41, MRG15, Eaf6, rept, Ing3, Tip60, Nipped-A, E(Pc), Act87E) highly enriched in our IP samples (Fig. 2b, dark blue). We also uncovered the Domino-interacting partner Arp6[34] (Fig. 2b, yellow dot) by IP-MS, highlighting the sensitivity of our approach.

Similar to H2A.Z, both Domino isoforms localized in the nucleus before and during ZGA, as shown by immunofluorescence (Fig. 2c and Supplementary Fig. 2a). But in contrast to H2A.Z, none of the Domino isoforms seem to remain attached to mitotic chromosomes (Supplementary Fig. 2bi). Although both

Domino isoforms showed cytoplasmic staining, this pattern was not similar to the small ring structures seen for H2A.Z on lipid droplets (Supplementary Fig. 2bii, compare to Supplementary Fig. 1e).

To determine whether maternally provided Domino accounts for the specific enrichment of H2A.Z on promoters in zygotes, we

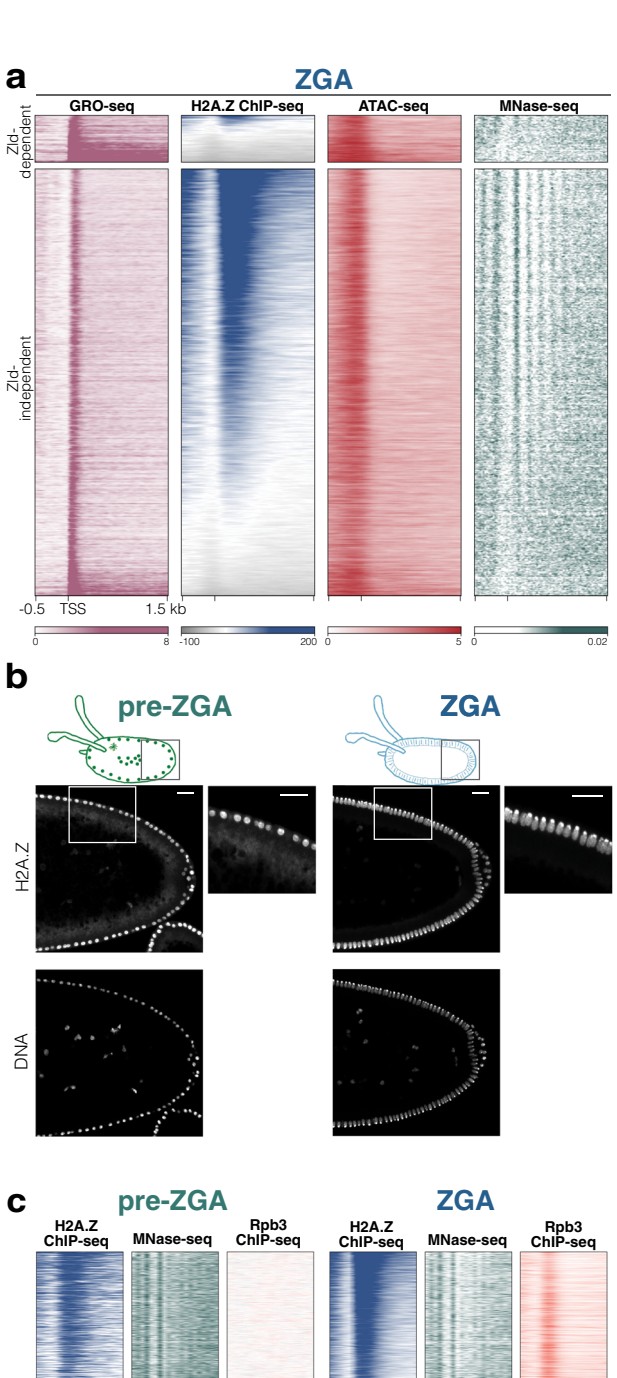

**Fig. 1 H2A.Z marks most of active TSS at ZGA and its deposition precedes ZGA. a** Heatmap of active promoters at NC14 determined by GRO-seq signal, and classified as Zld-dependent and Zld-independent according to ref. [10]. Sorting is based on H2A.Z signal and shows the distribution of H2A.Z (ChIP-seq), chromatin accessibility (ATAC-seq) and nucleosome positioning (MNase-seq) in these transcripts. Mean signal of three (GRO-seq) and two (H2A.Z ChIP-seq and ATAC-seq) replicates is shown. For MNase-seq, merged signal of three replicates from ref. [22] is shown (see Methods for details). See also Supplementary Fig. 1a, b. **b** Immunofluorescence of H2A.Z in pre-ZGA (left, NC11) and ZGA (right, NC14) embryos showing that H2A.Z localizes in the nucleus before and during ZGA. Insets show a magnification of the H2A.Z signal. Similar results were observed for at least 5 embryos of each stage. Scale bar = 20 μm. See also Supplementary Fig. 1e, f. **c** Heatmaps of transcripts with active promoters sorted by H2A.Z signal at ZGA. H2A.Z is already localized on chromatin before ZGA at NC9-12 (left, H2A.Z ChIP-seq) and nucleosomes are already positioned (left, MNase-seq), a lack of the Rpb3 signal (left) (Rpb3 ChIP-seq, representing RNA Pol II binding) confirms the pre-ZGA stage (see also Supplementary Fig. 1g, h). The same signal tracks are shown for embryos at ZGA for comparison (last 3 panels to the right). Mean signal of two replicates (H2A.Z and Rpb3 ChIP-seq) and merged signal of three replicates (MNase-seq, from ref. [22]) are shown per track.

depleted the Domino maternal supply by conducting RNAi-mediated knockdown of both Domino isoforms in late stage ovaries[20]. We confirmed that Domino knockdown (DomKD) embryos have reduced levels of Domino protein and mRNA by immunofluorescence, western blot and qPCR (Supplementary Fig. 2c–e). DomKD embryos showed no strong morphological defects at ZGA (Fig. 2d, left), but about 70% failed to complete embryogenesis (mean hatching rate for DomKD embryos 23.64% compared to 92.0% for Ctrl embryos) (Fig. 2d, right). Moreover, H2A.Z incorporation into chromatin was reduced by 73% in DomKD ZGA embryos compared to control embryos as measured by western blot (Supplementary Fig. 2f, g). We also observe this overall reduction of H2A.Z during mitosis upon DomKD by immunofluorescence (Supplementary Fig. 2h).

H2A.Z is the only H2A variant found in *Drosophila*. It belongs to the H2A.Z family based on its high sequence conservation[49]. However, it also contains an extended C-terminal domain with high resemblance to another eukaryotic and almost universal histone variant, called H2A.X. Both in *Drosophila* and other eukaryotes, this domain is essential for Double-Strand Break (DSB) repair via the phosphorylation of a serine residue[50]. Domino has already been directly implicated in the exchange of a phosphorylated form of H2A.Z for an unmodified form during DSB repair[45]. In order to see if the proportion of phosphorylated H2A.Z (phospho-H2A.Z) changed upon DomKD, we immunostained ZGA embryos with anti phospho-H2A.Z. Yet, we did not observe any radical change in the levels of phospho-H2A.Z (Supplementary Fig. 2i).

Quantitative ChIP[51] of manually staged ZGA embryos revealed a 43–57% (two biological replicates) reduction of H2A.Z at promoter regions in DomKD embryos compared to controls (Fig. 2e–g, see Methods). In contrast, genome-wide Zelda binding was not affected upon DomKD, as measured by quantitative Cut&Tag (Fig. 2f and Supplementary Fig. 2j). Surprisingly, we also did not see global changes in H4K12ac (Supplementary Fig. 2j), which has been associated with the DominoA (DomA) containing complex in Kc167 cells[34]. Reduced H2A.Z incorporation did not affect chromatin accessibility, as assessed by ATAC-seq (Supplementary Fig. 2k). Thus, maternally contributed Domino is necessary for the enrichment of H2A.Z at promoters

before ZGA onset, but does not affect Zelda binding, H4K12ac levels and chromatin accessibility.

**Domino knockdown reduces the transcription of housekeeping genes during ZGA.** Previous research showed that H2A.Z is a transcriptional regulator[29,32–35]. Given that Domino is required for H2A.Z deposition on promoters during early embryogenesis (Fig. 2e–g), we hypothesized that a lack of Domino, and thus a loss of H2A.Z enrichment, would cause transcriptional defects during ZGA. We focused our analysis only on genes with zygote-specific transcription (see Methods for details of maternal-zygotic classification and Supplementary Fig. 3a, b). We first performed RNA-seq on DomKD and control ZGA embryos and found that the transcript levels of Zld-dependent genes (lacking H2A.Z) showed no change upon loss of Domino (Supplementary Fig. 3c and Supplementary Data 3), whereas H2A.Z-positive genes showed the strongest reduction in transcript levels upon Domino knockdown (Supplementary Fig. 3c and Supplementary Data 3). To refine our approach and validate the RNA-seq results, we conducted GRO-seq in DomKD and control embryos during ZGA and performed differential gene expression (DGE) analysis. We found that H2A.Z-positive genes had significant down-regulation of transcription compared to H2A.Z-negative genes (Fig. 3a, Supplementary Data 3). Moreover, H2A.Z-positive loci exhibited the strongest reduction in Pol II occupancy in DomKD embryos compared to control embryos (Fig. 3b, Supplementary Data 3).

To better understand the roles of the different Domino isoforms in the transcriptional defects we performed a DominoB specific knockdown (DomB_KD). We also generated transgenes that bear a hairpin resistant variant of each Domino isoform and that we could overexpress in the DomKD background (see Methods). We validated each of these conditions by qPCR using domA and domB specific primers (Supplementary Fig. 3d).

DominoB specific knockdown embryos have a strong defect in completion of embryogenesis (mean hatching rate for DomB_KD embryos 30.41% compared to 96.39% for Ctrl embryos and 12.66% for DomKD embryos) (Supplementary Fig. 3e). Moreover, we found that the DominoB specific knockdown strongly reduced transcription of H2A.Z positive genes (Supplementary Fig. 3f). This suggests that DominoB is the major isoform responsible for the phenotype at this developmental stage and that the endogenous levels of the DominoA isoform cannot rescue the lack of DominoB.

Surprisingly, overexpression of a RNAi-resistant dominoA transgene fully rescued embryonic viability (mean hatching rate of 94.82%, Rescue DomA, Supplementary Fig. 3e), suggesting that in overexpression conditions DominoA isoform can compensate the lack of DominoB. Overexpression of an RNAi-resistant dominoB transgene also rescued embryonic viability (mean hatching rate of 53.33%, Rescue DomB, Supplementary Fig. 3e) but to a lesser extent in agreement with the lower levels of the overexpressed transcript (Supplementary Fig. 3d). Next, we checked by qPCR the transcriptional levels of two H2A.Z positive genes and one Zelda-dependent gene in each of the over-expression conditions. We found partial rescue of H2A.Z positive genes in both overexpression conditions (Supplementary Fig. 3f). Instead, although the Zld-dependent gene showed high variability, we did not see any strong downregulation. In summary, endogenous levels of DominoA are not enough to rescue the effect of DomB knockdown, however, the lack of Domino can be rescued by overexpressing any of the two isoforms.

Since we observed similar phenotypes and transcriptional defects between our DomKD (both isoforms) and a DomB specific knockdown (DomB_KD), we decided to inspect other

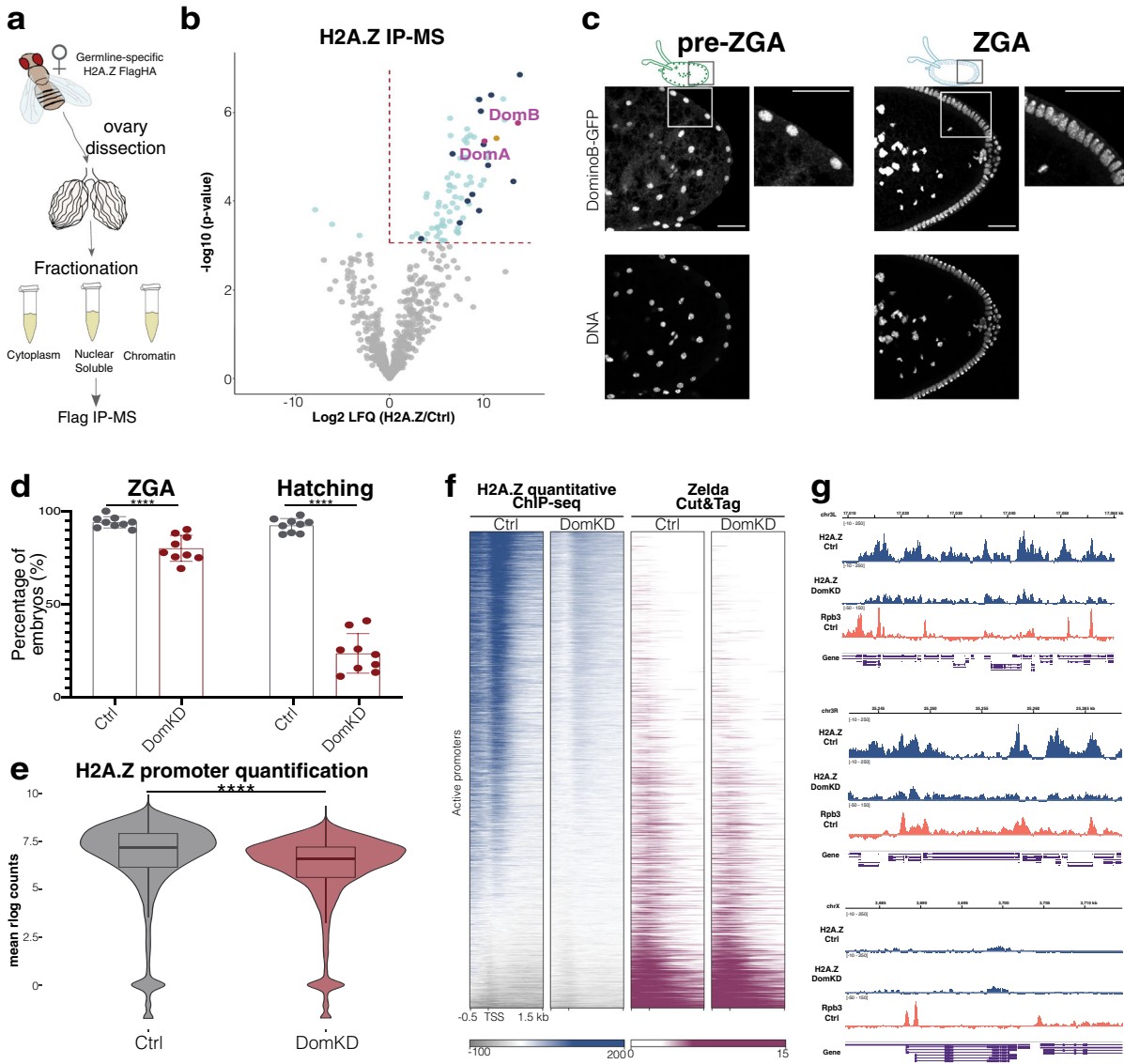

**Fig. 2 Domino is the main histone chaperone for H2A.Z on TSS. a** Experimental setup for H2A.Z IP-MS. **b** Volcano plot showing the -log10 p-value (y axis) compared to the enrichment over control (log2 LFQ) of the anti-Flag IP followed by MS. A fitted linear model with padj for multiple-hypothesis (Benjamini–Yekutieli, two-sided) was used to obtain significantly enriched or depleted proteins, colored in cyan (-log10 padj value >1.3). The red dotted line frames significantly enriched proteins (log2 LFQ > 0). Domino isoforms are highlighted in magenta. Dark blue denotes significantly enriched members of the Tip60 complex. Yellow is Arp6, a DominoB[34] interactor. $n = 3$ biological replicates per condition. **c** Representative immunofluorescence of DominoB-GFP in pre-ZGA (left, NC10) and ZGA (right, NC14) embryos. Insets show a magnification of the GFP signal. Scale bar = 40 μm. See Supplementary Fig. 2a for DomA staining. **d** Phenotypic characterization of Ctrl and DomKD embryos. Percentage of embryos (left) reaching the ZGA stage or (right) completing embryogenesis (Hatching) (p-value < 0.0001, two-tailed Mann–Whitney test). $n = 453$ embryos for Ctrl and $n = 465$ embryos for DomKD embryos. Data are presented as mean values ± SD. Source data are provided as a Source Data file. **e** Violin plot of the mean rlog transformed counts of H2A.Z quantitative ChIP-seq on active promoter regions in Ctrl and DomKD embryos at ZGA. Box plot inside depict the median and the interquartile range (IQR) from the 1st to the 3rd quartile. Whiskers indicate the upper (Q3 + 1.5*IQR) and lower edge (Q1 − 1.5*IQR). Welch two sample t-test, two-sided, p-value < 2.2e−16. $n = 2$ biologically independent experiments per genotype. **f** Heatmap of H2A.Z quantitative ChIP-seq signal (left 2 columns) and Zelda Cut&Tag signal (right 2 columns) on active promoters, sorted according to H2A.Z signal on Ctrl embryos at ZGA. Averaged library size corrected signal tracks from two replicates were used for Zelda. **g** Screenshots of genome browser tracks (Integrative Genomics Viewer, IGV)[114] showing H2A.Z quantitative ChIP-seq signal in Ctrl and DomKD embryos at ZGA, Rpb3 signal in Ctrl embryos represents RNA Pol II coverage on the same regions. (Top) Chr3L:17,010,000–17,060,000. (Middle) Chr3R: 25,243,000–25,265,000. (Bottom) ChrX: 3,680,000–3,715,000.

interaction partners. The protein Host cell factor (Hcf) has been found to interact more strongly with DominoB than with DominoA in Kc167 cells[34]. Knocking down Hcf had a strong effect on the completion of embryogenesis (Supplementary Fig. 3g, mean hatching rate 2.22% compared to 91.15% in Ctrl embryos) and also showed a trend towards transcriptional downregulation of H2A.Z positive genes at ZGA, though not as

strong as for the DomKD background (Supplementary Fig. 3h). This data suggests that at least the DominoB-containing complex is involved in the transcriptional regulation of H2A.Z positive genes.

Given that H2A.Z-negative genes also showed a slight reduction in transcription (Fig. 3a) and Pol II occupancy (Fig. 3b), we investigated this phenomenon more in depth.

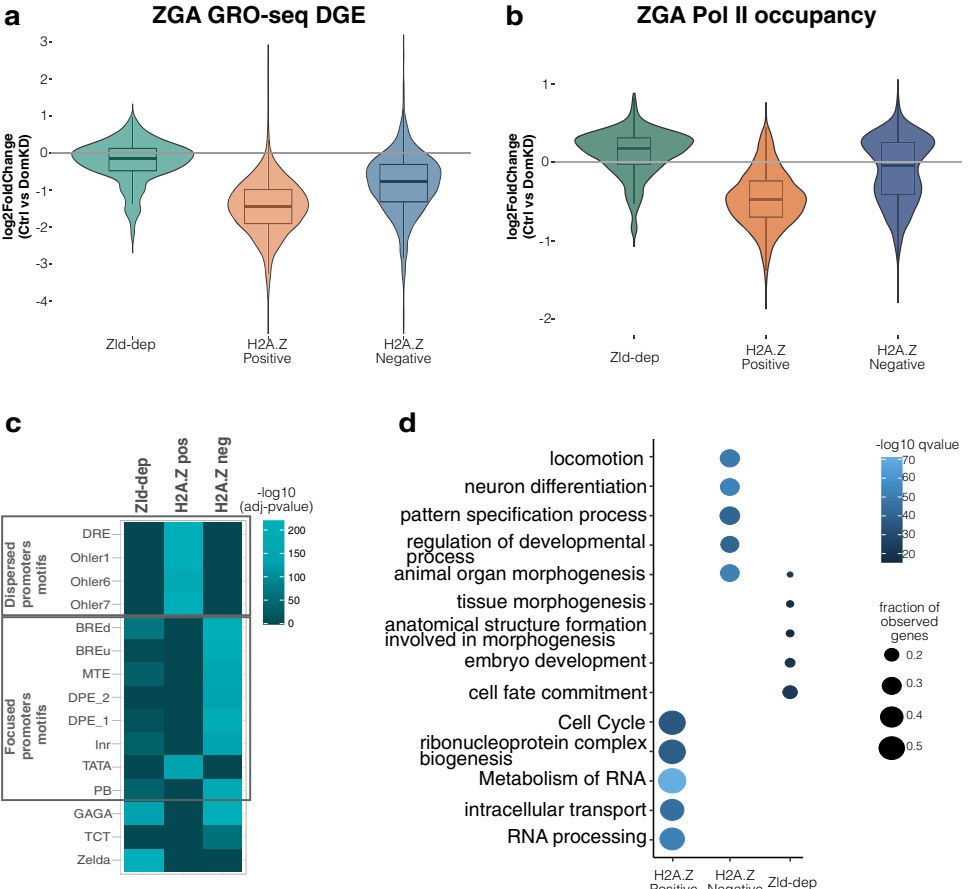

**Fig. 3 Transcription of housekeeping genes is affected in DomKD embryos. a** Violin plot showing the differential gene expression (DGE) as log2FoldChanges in gene expression of Ctrl vs DomKD in Zld-dependent, H2A.Z positive and H2A.Z negative genes measured by GRO-seq. See also Supplementary Fig. 3j. $n = 3$ biologically independent experiments per genotype. **b** Violin plot of the log2FoldChange for Pol II occupancy (Rpb3 ChIP-seq) per unique promoter of Zld-dependent, H2A.Z positive and H2A.Z negative promoters. See also Supplementary Fig. 3i. $n = 2$ biologically independent experiments per genotype. **a**, **b** All groups are significantly different by pairwise comparison, Wilcoxon rank sum test with continuity correction, two-sided, $p$-value < 2.2e−16. Box plots (**a**, **b**) depict the median and the interquartile range (IQR) from the 1st to the 3rd quartile. Whiskers indicate the upper ($Q3 + 1.5*IQR$) and lower edge ($Q1 − 1.5*IQR$). **c** Motif enrichment analysis of core promoter elements in Zld-dependent (Zld-dep), H2A.Z positive (H2A.Z pos) and H2A.Z negative (H2A.Z neg) promoters. **d** Gene set enrichment analysis using Metascape[56] showing the top 5 most significant GO terms per group (Zld-dep, H2A.Z pos and H2A.Z neg). Each GO-term is supported by the color scale in -log10 $q$-value and the fraction of observed genes presenting the proportion of observed genes in the respective GO term. Data representation was arranged as proposed by ref. [115]. See also Supplementary Fig. 3l. **c**, **d** Source data are provided as a Source Data file.

Although H2A.Z-negative genes do not contain H2A.Z at the +1 nucleosome position, 26% of them contain H2A.Z on the −1 nucleosome position. We observed that H2A.Z-negative genes containing H2A.Z in the −1 position showed reduction in Pol II occupancy compared to those that do not contain the variant (Supplementary Fig. 3i). We observed the same trend in the GRO-seq differential gene expression, where H2A.Z negative genes containing H2A.Z on the −1 nucleosome position showed reduction in transcription compared to H2A.Z negative genes that do not contain the variant at all (Supplementary Fig. 3j, compare H2A.Z negative "−1 nucleosome" versus "None").

These transcriptional defects in DomKD embryos prompted us to investigate whether H2A.Z-positive genes share specific core promoter elements, when compared to Zld-dependent genes and H2A.Z-negative genes. Core promoter elements are sequence motifs bound by basal transcription factors[52]. Typically located within 100 bps of the TSS, core promoter elements primarily drive two different transcription initiation modes: a focused mode, in which the TSS spans a narrow window of a single or a few (<5) nucleotides, and a dispersed mode, where many TSSs are spread over a wide window, which in *Drosophila* can be up to

100 bp[53]. We found that H2A.Z-positive genes are enriched for distinct core promoter elements, such as Ohler1, Ohler6, Ohler7, and DRE motifs (Fig. 3c)[52–54], which are characteristic of dispersed promoters. In contrast, the promoters of Zld-dependent genes are enriched for the Zelda (also known as TAGteam) motif and focused promoter elements, such as BRE and Inr, in agreement with previous studies[55]. Notably, the promoters of H2A.Z-negative genes also contain sequence motifs such as DPE, MTE and the BRE motifs, which are associated with focused promoters (Fig. 3c). We also found that these three groups (Zld-dependent, H2A.Z positive and H2A.Z negative promoters) also contain different sets of transcription factor motifs (Supplementary Fig. 3k), which further strengthens the idea of different regulatory mechanisms.

To confirm these findings, we performed a Gene Ontology (GO) analysis using Metascape[56] (Fig. 3d and Supplementary Fig. 3l). We observed that H2A.Z-positive genes are associated with basic cellular processes such as cell cycle and RNA metabolism, whereas both H2A.Z-negative and Zld-dependent genes show a strong trend towards developmental specific functions such as neuron differentiation or tissue morphogenesis.

This finding was further corroborated by analyzing gene expression over developmental time using data produced for the modENCODE project[57]. Here, we found that H2A.Z-positive genes are transcribed continuously throughout embryonic development and at relatively stable levels (a characteristic of housekeeping genes), whereas H2A.Z-negative and Zld-dependent genes exhibit peaks of expression at different developmental stages, which results in a higher variance of expression through time (a feature of developmentally regulated genes) (Supplementary Fig. 3m, n). Moreover, we found that 67.6% of genes already reported as housekeeping by ref. [58] where within our H2A.Z positive genes. Overall, these results indicate that H2A.Z deposited by maternally supplied Domino controls the transcriptional activation of housekeeping genes during ZGA.

Given that H2A.Z was also found in some inactive genes (Supplementary Fig. 1a) we wanted to investigate whether H2A.Z also functions as a marker for future expression. Using modENCODE data[57], we checked the expression of the inactive genes through development, splitting them first by origin (maternal or non-maternal) and then by H2A.Z content (H2A.Z positive and H2A.Z negative). We did not see any difference among the non-maternal genes, regardless of the content of H2A.Z (Supplementary Fig. 3o, non-maternal). Nevertheless, for those that are maternally deposited we observed an increase in expression compared to the H2A.Z negative genes of the same condition. This increase starts shortly after ZGA (4–6 h) and becomes more prominent towards the 10 h of development (Supplementary Fig. 3o, maternal). We hypothesized that the presence of H2A.Z could facilitate the zygotic re-expression of genes expressed in the maternal germline.

**Nucleosome positioning occurs independently of Domino and H2A.Z.** Chromatin remodeling complexes change the composition, packaging, and positioning of nucleosomes, thereby regulating chromatin structure and accessibility[59]. While our results show that H2A.Z deposition correlates with well-positioned nucleosomes (Fig. 1a), it is unclear whether Domino and H2A.Z directly influence nucleosome positioning. To test this hypothesis, we conducted MNase-seq on DomKD embryos during ZGA. We found no significant differences in nucleosomal positioning between DomKD and control embryos in either Zld-dependent, H2A.Z-positive or H2A.Z-negative genes (Fig. 4a and Supplementary Fig. 4a), showing that nucleosome positioning occurs independently of H2A.Z deposition. These results are consistent with our ATAC-seq analysis, which revealed no differences in open chromatin regions between DomKD and control embryos (Supplementary Fig. 2k). Together, these results suggest that nucleosome position and chromatin accessibility are established upstream and independently of Domino and H2A.Z.

**Domino is required for the correct establishment of TAD boundaries at ZGA.** During ZGA, 3D chromatin structure elements such as TADs become established. We found that Zld-dependent, H2A.Z-positive and H2A.Z-negative loci have characteristic insulation scores (Supplementary Fig. 1c). To investigate whether H2A.Z is involved in establishing TAD structures, we performed HiC on DomKD and control embryos during ZGA. Whole genome analyses showed that Domino knockdown does not affect chromatin conformation or the appearance of TADs (Supplementary Fig. 4a, b). However, when we examined each gene category separately by measuring insulation score, H2A.Z-positive genes displayed the strongest loss of insulation at TSS in DomKD embryos compared to controls (Fig. 4b, c). This was confirmed by checking for the drop of insulation scores sorted by H2A.Z ChIP-seq signal (Fig. 4c). Thus, our data show that

Domino is required for the local definition of TAD insulation at H2A.Z-positive genes.

It has been reported that a knockdown of Zelda reduces the insulation score in TAD boundaries with the strongest Zelda binding[23]. We asked whether insulation scores were also decreased in H2A.Z positive loci upon Zelda knockdown (ZldKD). For that, we check for H2A.Z positive loci in ZldKD embryos using data from Hug et al., 2017[23]. In accordance with their results, ZldKD caused a decrease in insulation score at Zld-dependent promoters but not in H2A.Z positive and H2A.Z negative genes (Supplementary Fig. 4c, d). Surprisingly we found a slight increase of insulation in both H2A.Z positive and H2A.Z negative genes upon ZldKD that we speculate is an indirect effect due to the loss of insulation in Zld-dependent promoters.

## Discussion

This study uncovers a role for the histone variant H2A.Z as an important regulator of gene activation and transcription during ZGA. Our results demonstrate that H2A.Z is deposited at the TSS of the zygotic genome by maternally supplied Domino, an ATPase and histone chaperone of H2A.Z. Moreover, we show that H2A.Z deposition precedes ZGA and also precedes Pol II binding to chromatin (Fig. 1c), suggesting that it primes genes for transcriptional activation. Additionally, H4K12 acetylation (a mark recently shown to be dependent on DominoA[34]) was not affected in DomKD embryos at this developmental stage (Supplementary Fig. 2f, g), further suggesting that all transcriptional defects at ZGA were caused specifically by the absence of H2A.Z in promoter regions.

Surprisingly, our results showed that absence of H2A.Z in promoter regions does not affect chromatin accessibility or nucleosome positioning, indicating that the regulation of these two processes in the early embryo is independent of Domino/H2A.Z deposition. Several factors influence in vivo nucleosome positions, including the DNA sequence, transcription factors, the Pol II complex or ATP dependent remodelers[59–61]. ATP dependent chromatin remodelers can modify the position of nucleosomes either by nucleosome sliding, or by nucleosome eviction/incorporation[59]. As mentioned previously, the nucleosome free region (NFR) and the well-positioned +1 nucleosome are maintained upon DomKD, suggesting that additional ATPases maintain the well-positioned nucleosomes.

We observed a reduction in Pol II occupancy following knockdown of Domino which may be explained by the requirement for either Domino or H2A.Z during Pol II recruitment. In this context, it is notable that both H2A.Z and Domino interact with transcription factors[62,63] and in yeast, the Domino ortholog SWR1 associates with NFR surrounding the TSS[64]. Additionally, we found that H2A.Z positive genes have a characteristic promoter structure of housekeeping genes (Fig. 3c). Interestingly, the transcription factor motifs associated are also substantially different from both Zld-dependent and H2A.Z negative genes (Supplementary Fig. 3k). From that, we speculate that H2A.Z positive genes have an entirely different regulation, that goes from the mode of transcription initiation to the binding of transcription factors. Future research on this direction will allow us to identify the precise factors that guide Pol II recruitment by Domino or H2A.Z at this specific developmental stage.

Depletion of Pol II on H2A.Z positive genes could also explain the decrease on TAD insulation upon DomKD. A previous study showed that treatment with transcription-inhibitor drugs like triptolide or alpha-amanitin have a similar effect on insulation[23]. Moreover, Zld knockdown causes a loss of insulation only at Zld-dependent genes[23], which also show depletion of Pol II occupancy[10].

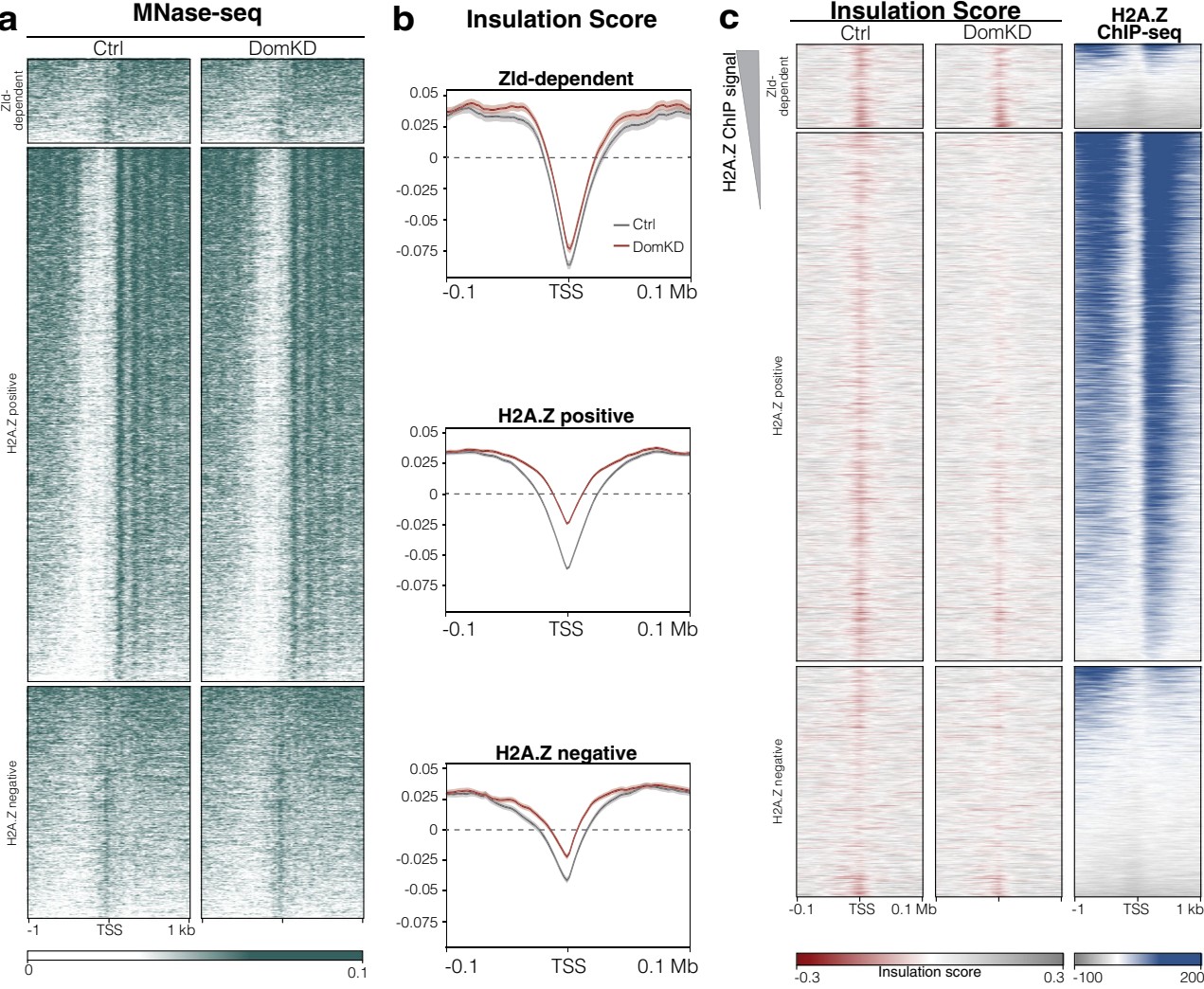

**Fig. 4 Domino is required for the correct establishment of TADs insulation but not for nucleosome positioning. a** Heatmap of MNase signal in Ctrl and DomKD embryos at ZGA in Zld-dependent, H2A.Z positive and H2A.Z negative promoters. The heatmap is sorted by MNase-seq signal in Ctrl embryos. Tracks show the signal obtained after merging two biological replicates per condition. **b** Profile plots of insulation score in Zld-dependent, H2A.Z positive and H2A.Z negative promoters comparing Ctrl vs DomKD embryos at ZGA. The center line indicates the median and the semi-transparent lines represent the standard error of a given group and condition, which indicates that in all groups DomKD is significantly different than the control at TSS. **c** Heatmap of the insulation score in Ctrl and DomKD embryos sorted by H2A.Z Ctrl signal at ZGA. All tracks are from embryos at ZGA.

The Domino orthologs in mammals, SRCAP and EP400, have not been studied during early embryogenesis. EP400 has been shown to be essential for ESC identity[65] and *EP400* mutant mice are homozygous lethal[66]. To our knowledge, there are no reports on the viability of homozygous mutants for the mouse *Srcap* gene. In humans, hemizygous mutations producing a truncated version of SRCAP are associated with Floating Harbor Syndrome[67]. Recently, a mutation in SRCAP has been associated with uterine leiomyoma[68]. However, the potential function of SRCAP in transcription regulation has not been investigated.

H2A.Z is conserved from yeast to mammals and is essential for viability in *Xenopus*, *Drosophila* and mouse[36–38]. The storage of H2A.Z in lipid droplets through the protein Jabba has been shown to be important in the regulation of its nuclear levels[39,40,69]. However, its effects on the chromatin bound fraction are unknown. The presence of H2A.Z as a marker for future activation has been already suggested in yeast, where H2A.Z has shown to be important for the transcriptional activation of repressed or lowly expressed genes[70]. In the same way, in mouse embryonic stem cells (mESCs), H2A.Z on chromatin correlates

with H3K4me3, contained both in active promoters and bivalent promoters (that also contain H3K27me3), but not in repressed genes; moreover, this pattern is conserved also in human embryonic stem cells (hESCs)[32]. Similarly, in zebrafish embryos the histone variant H2A.Z is inherited from the paternal germline and is retained on promoters that would become activated or bivalent[19].

Zelda is a pioneer factor essential for ZGA. Lack of Zelda stops development and causes embryos to die just before ZGA[7]. However, this factor controls only about 10% of the genes activated in the early embryo, suggesting that additional regulators are involved in this process. Along this line, other transcription factors, such as *Odd pair* (Opa) and *GAGA Factor* (GAF), have been also identified as important during ZGA[71–73]. We show evidence that H2A.Z deposited by maternally provided Domino is required for the transcriptional activation of thousands of genes at ZGA onset. This finding broadens our current understanding of ZGA regulation by highlighting the importance of chromatin in regulating this process. Given the evolutionary conservation of H2A.Z and of the fundamental principles underlying ZGA, we

speculate that histone variants could have similar functions during mammalian embryogenesis. Future research on this path will help to shed light on the complex process by which chromatin states and transcription factors together orchestrate zygotic genome activation.

## Methods

**Fly stocks**. All the stocks used in this study were grown on corn flour molasses food (12 g agar, 18 g dry yeast, 10 g soya powder, 22 g molasses, 80 g malt extract, 80 g corn flour, 6.25 ml propionic acid, 2.4 g nipagin per liter of water) at 25 °C. Fly stocks used in this study: TRiP line control (BDSC#36303), Domino shRNA (BDSC #41674), DominoB shRNA (BDSC #55917), Hcf shRNA (BDSC #36799), pUASp H2Av::FlagHA (This study), mat-alpha-Gal4 (BDSC #7062), DomA-GFP-Flag (kindly provided by Peter Becker, LMU Munich), DomB-GFP-Flag (kindly provided by Peter Becker, LMU Munich), deltaDom-GFP (kindly provided by Peter Becker, LMU Munich), UASp-DomB-Flag-HA (Rescue DomB,this study), UASp-DomA-Flag-HA (Rescue DomA, this study).

**Maternal RNA depletion**. Maternal RNAi knockdown (KD) was induced as specified in ref. [20]. Briefly, UAS-shRNA (TRiP line) females were crossed with germline-specific Gal4-driver (mat-alpha-Gal4, BDSC#7062) carrying males. The progeny of this crosses (F1 generation) is KD for the specific target in the germline and will lay eggs that are depleted of this particular transcript.

**Embryo phenotypic characterization**. Embryo hatching rate was performed as in ref. [20]; together with the quantification for embryos reaching ZGA. Briefly, a minimum of 50 embryos were randomly picked. An embryo was considered as cellularized when it displayed normal morphological features of stage 5 (ZGA, NC14), which are a clear rim on the embryo periphery and the complete invagination of cellular membranes. Number of hatched embryos was counted after 26hrs. n represents the total number of embryos used for the full experiment.

**Embryo collection**. Unfertilized eggs and embryos for RNA-seq and ChIP-seq experiments were collected as in ref. [20]. Embryos for ATAC-seq, Cut&Tag and ChIP-seq were collected and fixed with 1%PFA as specified in ref. [20]. For GRO-seq, RELACS ChIP-seq, HiC and MNase-seq experiments, embryos were hand-picked using halocarbon oil 27 (Sigma 9002-83-9) and a stereoscope equipped with transmitted light. For GRO-seq experiments, embryos were collected in batches of 100 and shock-frozen. Five hundred embryos (5 batches of 100 embryos) were pooled for each replicate. For MNase-seq experiments, 100 embryos per replicate were hand-picked, placed in 50 μl of wash buffer (20 mM HEPES[pH7.5], 150 mM NaCl, 0.5 mM Spermidine, 1× Protease inhibitors), and washed twice with wash buffer. For RELACS ChIP-seq and HiC experiments, embryos were dechorionated in 50% bleach for 2 min, transferred to 7 mL heptane and crosslinked with 5 mL of fixative. The fixative consisted of bufferA (60 mM KCl, 15 mM NaCl, 15 mM HEPES [pH 7.6], 4 mM MgCl$_2$) supplemented with 1.8% PFA (for RELACS ChIP-seq and HiC experiments). Embryos were fixed for 15 min in an orbital shaker at maximum speed. The crosslinking was quenched by adding a final concentration of 225 mM glycine and rotation for 5 min. After washing with bufferA + 0.1% Triton X100, the embryos were hand-staged on a cooling station under a microscope, shock frozen and stored at −80 °C. For each RELACS ChIP-seq replicate, 400 ZGA embryos were used. Each HiC replicate and ATAC-seq replicate used 100 and 25 ZGA embryos, respectively.

**GRO-seq**. Three biological replicates were performed per genotype. The GRO-seq protocol was adapted from ref. [74] with modifications in the nuclei extraction protocol as follows: Embryos were resuspended in 1.5 ml nuclei extraction buffer (3 mM CaCl$_2$, 2 mM MgCl$_2$, 10 mM Tris HCl pH 7.5, 0.25% Np40, 10% Glycerol, Protease inhibitors and RNase inhibitor 4U/ml) and stroked 60 times. The lysate was centrifuged at 100 × g and nuclei were pelleted at 1000 × g, and washed afterwards four times with nuclei extraction buffer. Nuclei were washed once with freezing buffer (50 mM Tris HCl pH 8, 5 mM MgCl$_2$, 0.1 mM EDTA) and resuspended in 100 μl of freezing buffer. The Nuclear Run-On (NRO) reaction was performed by addition of 100 μl NRO 2× buffer (10 mM Tris HCl, 5 mM MgCl$_2$, 1 mM DTT, 300 mM KCl, 1% Sarkosyl, 0.5 mM ATP, CTP and GTP and 0.8 U/μl RNase inhibitor), using 1 mM Bio-11-UTP final concentration, followed by an incubation of 5 min at 30 °C. RNA extraction was performed following the manufacturer's instructions with TRIzol reagent. Purified RNA was fragmented by the addition of reverse transcriptase buffer and 7 min incubation at 95 °C. Biotinylated nascent RNAs were bound to 30 μl Dynabeads MyOne Streptavidin C1 (Invitrogen), washed and purified. Fragmented RNAs were repaired incubating with Polynucleotide kinase (Thermo scientific) for 30 min at 37 °C. RNA was purified with Phenol:Chloroform and precipitated. RNA was then ligated to 3' end adapter using T4 RNA ligase 2 Truncated KQ (home-made) for 16 h at 15 °C. After ligation RNA was purified using SPRI beads and biotinylated RNA was enriched as described above. Purified RNA was ligated at 5' end using T4 RNA ligase1 for 2 h at 25 °C. Then, RNA was purified using SPRI beads and biotinylated RNA was

enriched for a third time as described above. Reverse transcription was performed using SuperScript IV Reverse Transcriptase (Thermo Fisher Scientific). The RT reaction was incubated for 1 h at 50 °C. cDNA was amplified with specific primers using Phusion High fidelity PCR master mix 2× (New England Biolab) and sequenced on Illumina NextSeq 500 system.

**ChIP-seq**. Two biological replicates were performed for each ChIP-seq experiment. ChIP-seq experiments were performed as in ref. [20]. Briefly, nuclei were extracted in lysis buffer (140 mM NaCl, 15 mM HEPES [pH 7.6], 1 mM EDTA, 0.5 mM EGTA, 1% TritonX100, 0.5 mM DTT, 0.1% Sodium Deoxycholate, 10 mM Sodium Butyrate, 1× Protease Inhibitors) and subsequently subjected to ultrasound treatment (Covaris E220, 45 s, peak power 75, duty factor 10, cycles burst 200). Fixed chromatin was then sheared using Covaris E220 (900 s, peak power 140, duty factor 5, cycle burst 200) and precleared overnight. Each sample was incubated with the relevant antibody and concentration (See Supplementary Table 1) for at least 4 h at 4 °C. Samples were then washed, eluted and decrosslinked overnight by incubation at 65 °C. Next, samples were treated with RNaseA (50 μg/mL final concentration) for 30 min at 37 °C and ProteinaseK (200 μg/mL, final) for 3 h at 56 °C. DNA was purified and libraries were prepared according to manufacturer's instructions using the NEBNext Ultra II DNA Library Prep Kit for Illumina. Libraries were quality controlled by capillary electrophoresis on the Fragment Analyzer system (Advanced Analytical). Sequencing was done using the HiSeq, NextSeq or NovaSeq Illumina platform together with the paired end sequencing option.

**RELACS ChIP-seq**. Two biological replicates were done per genotype. RELACS protocol was performed as described previously in ref. [51]. Briefly, embryos were thawed in RELACS lysis buffer (10 mM Tris-HCl [pH 8], 10 mM NaCl, 0.2% Igepal, 1× Protease inhibitor cocktail) and the nuclei were isolated by sonication using the NEXSON procedure[75]. To digest the chromatin, 25 μl of 10× CutSmart buffer (NEB), 2.5 μl 100× Protease inhibitor cocktail and 1 μl of CviKI-1 (50 U/μl, NEB R0710S) were added. The digestion reaction was incubated overnight at 20 °C. End repair and A-tailing was performed and customized adapters[51] were ligated to the fragments. Once barcoded, the samples were pooled together. Chromatin was then sheared by sonication (Covaris E220, MicroTubes, 5 min, peak power 105, duty factor 2, cycles burst 200). This chromatin was used for automated ChIP with the IP-Star Diagenode system. IPs and Inputs were decrosslinked, DNA was purified and libraries were prepared according the NEB Ultra II DNA Library Prep Kit for Illumina (E7645S and E6440) following the manufacturer's instructions. Integrity and size-distribution of the samples was assessed before and after library preparation by running on Fragment Analyzer (Advanced Analytical). See Supplementary Table 1 for a list of antibodies used.

**Cut&Tag**. Cut&Tag experiments were performed as in ref. [76]. Embryo collection and fixation was performed as stated in the Embryo collection section. 50 hand-staged embryos were used per sample. DNA was purified and 1 pg of *lambda* phage genome (previously treated with ProteinA/G-Tn5) was added to each sample as spike-in before library preparation. See Supplementary Table 1 for a list of antibodies used.

**HiC**. For each genotype, two biological replicates were analyzed. 50 μl ice-cold HiC lysis buffer (10 mM Tris-HCl [pH 8], 10 mM NaCl, 0.2% Igepal, 1× Protease Inhibitor Cocktail (Roche, 11836170001)) were added and embryos were crushed with a pestle. Then, nuclei were pelleted by centrifugation (3000 × g, 5 min at 4 °C). Next, the pellet was incubated in 50 μl 0.5% SDS for 10 min at room temperature. The reaction was quenched by adding 145 μl of water and 25 μl of 10% Triton X100. For chromatin digestion, 0.7 μl of DpnII (NEB R0543T) and 25 μl of DpnII buffer were added. The reaction was incubated for 90 min at 37 °C at 600 rpm and a second instance of 0.7 μl of DpnII was added followed by a second incubation of 90 min. Nuclei were then pelleted and resuspended in 135 μl of 1× CutSmart buffer (NEB B7204S). To biotinylate the free chromatin ends, 0.15 mM (final) of dNTP-mix (NEB), 0.05 mM (final) of Biotin-14-dATP (Jena Bioscience, NU-835-BIO14-S) and 10 U of Klenow (NEB M0210 L) was added to the reaction and incubated for 60 min at 25 °C. The chromatin free ends were then ligated by adding ligation buffer (NEB B0202S) (1× final concentration), TritonX-100 (0.8% final concentration), 120 μg BSA and 2000 U ligase (NEB M0202S). This reaction was incubated for 2 h, after which another 2000 U of ligase were added and incubated for another 2hrs. Nuclei were then pelleted and resuspended in 200 μl elution buffer (10 mM Tris-HCl [pH 8], 1 mM EDTA, 1% SDS). Proteins were digested upon addition of 400 μg Proteinase K for 30 min at 55 °C. The chromatin was reverse crosslinked in the presence of 0.365 M NaCl at 68 °C overnight. DNA was purified using the ChIP DNA Clean&Concentrator Kit (Zymo Research D5205) eluted in 50 μl and biotin was removed from non-ligated fragments by adding NEB buffer 2 (NEB B7002S) (1× final), 12 μg of BSA, dATP and dGTP (0.025 final each), 3 U T4 DNA polymerase (NEB M0203S) in a final volume of 120 μl. The reaction was stopped by adding a final concentration of 13 mM EDTA. Samples were sheared with Covaris E220 (Covaris microTUBE snap cap (520045) with intensifier (50014), duty factor 10%, peak incident power 140, cycle/burst 200, 120 s) to generate fragments of 200–300 bp. Ligated and biotinylated fragments were enriched through biotin pulldown using Dynabeads MyOne Streptavidin C1

(Invitrogen 65001). Afterwards, 2× binding buffer (10 mM Tris-HCl [pH 8], 1 mM EDTA, 2 M NaCl) (1× final concentration) was added to each sample. Subsequently, beads (equilibrated with TWB (5 mM Tris-HCl [pH 8], 0.5 mM EDTA, 1 M NaCl, 0.05% Tween) resuspended in 1× binding buffer were added to each sample. The beads were washed twice for 2 min at 55 °C with TWB, twice with EB-buffer (Qiagen), resuspended in 50 µl of EB-buffer and directly used for library preparation using the NEB Ultra II DNA Library Prep Kit for Illumina (E7645S and E6440) following the manufacturer's instructions except that the adapter ligation occurred on the biotin beads. The fragments were released from the beads before the PCR reaction (by heating to 98 °C for 10 min) once non-ligated adapters had been removed.

**Embryo immunofluorescence**. For H2A.Z immunostaining, embryos were dechorionated and heat fixed as follows: 500 µl of boiling Triton-X Salt solution (0.03% TritonX100, 0.068 M NaCl) was added immediately followed by addition of 500 µl of ice-cold Triton-X Salt solution and 5 min incubation on ice. After removing this solution, 500 µl of Heptane and 500 µl of MeOH were added and vigorously shaken. The embryos were then washed 3 times with MeOH and stored. For Domino-GFP immunostaining, embryos were fixed as specified in ref. [20]. Immunostaining was performed as in ref. [20]. All images were acquired using the confocal laser scanning microscopes Leica TCS SP5, Zeiss LSM880-Airyscan or Zeiss ELYRA PS1. Stacks were assembled using Imaris 9.5.1 (Bitplane). For all antibodies used see Supplementary Table 1.

**Ovary fractionation**. For each replicate, 100 pairs of ovaries were dissected in 1× PBS. After removing the 1× PBS, ovaries were resuspended in 1 ml hypotonic buffer (15 mM HEPES [pH8.0], 350 mM Sucrose, 5 mM MgCl2, 10 mM KCl, 0.1 mM EDTA, 0.5 mM EGTA, 10 mM Beta-Mercaptoethanol, 0.2 mM PMSF, 1× Protease inhibitors) and dounced with 15 strokes. Nuclei were incubated on ice for 15 min and centrifuged ($9000 \times g$, 4 °C for 15 min). Samples were then washed (1 ml hypotonic buffer) and centrifuged ($9000 \times g$, 4 °C for 15 min). The pellet was resuspended in 500 µl of Mix-Salt Buffer (20 mM HEPES [pH7.9], 25% glycerol, 1.5 mM MgCl2, 400 mM KCl, 0.2 mM EDTA, 0.2 mM PMSF, 0.5 mM DTT, 1× Protease Inhibitors). Samples were incubated for 15 min on ice. Subsequently, the chromatin fraction was pelleted by centrifugation ($20,000 \times g$, 4 °C for 15 min). The supernatant (containing the nuclear soluble fraction) and the pellet (chromatin fraction) were separated. Protein concentration was measured using the Bradford assay. The fractionation quality was checked by Western Blot. The nuclear soluble fraction was then used for affinity purification using the Flag tag.

**Affinity purification**. Three biological replicates were done for H2A.Z FlagHA affinity purification. 25 µl of EZview™ Red ANTI-FLAG™ M2 Affinity Gel (Sigma, F2426) beads per sample were washed and equilibrated by adding first one bead volume of 0.1 M glycine [pH2.5], incubated for 3 min at room temperature and then adding 1 ml of 1 M Tris-HCl [pH7.9]. Beads were then washed twice with 500 µl IP-buffer (20 mM Tris-HCl [pH7.5], 10% glycerol, 5 mM MgCl2, 150 mM KCl, 0.05% NP40, 0.1% Tween20, 1 mM DTT, 1× Protease inhibitor cocktail). Ultimately, IP-buffer was added reconstituting the original 25 µl volume per sample.

Afterwards, 25 µl of beads was added to 1 mg of nuclear soluble extract in 400 µl IP-buffer, per sample. This solution was incubated for 4 h at 4 °C on a rotating wheel. Beads were then washed three times by adding 1 ml of IP-buffer and centrifugation ($1000 \times g$, 1 min at 4 °C). For elution, three bead volumes of IP-buffer with Flag peptide (200 ng/µl, final concentration) were added to the beads, incubated for 1 h at 4 °C and centrifuged ($1000 \times g$, 1 min at 4 °C). A second elution was then performed by repeating the previous steps. Both elutions were cleared using Pierce columns (ThermoFisher, 69702). The affinity purification quality was checked for the enrichment of the tagged protein in both elutions over the input control by SDS-PAGE analysis. Only afterwards, the two elutions were pooled together for further processing.

**Mass spectrometry sample preparation**. Bait and control pull-downs were prepared pursuing the paramagnetic bead-based single-pot, solid-phase-enhanced sample-preparation (SP3) method[77]. Briefly, pull-downs were buffered and adjusted to 50 µl volume in 100 mM Tris-HCl (pH 7.5), reduced (DTT; 12 mM final concentration) at 45 °C for 30 min, followed by cysteine alkylation (2-iodoacetamide; 40 mM final concentration) for 30 min at room temperature (RT) in the dark. 100 µg SP3 beads (1:1 mixture of GE Life Sciences carboxylate-modified Sera-Mag Speed Beads A (hydrophilic) and B (hydrophobic), respectively) were added together with acetonitrile (70% final concentration) and incubated on a shaker for 15 min at RT. In all further steps, magnetic beads were collected on an in-house fabricated Neodymium (Supermagnete, Germany) magnetic rack. Beads suspension was washed twice with ethanol (70%) and once with 100% acetonitrile. Proteolytic digestion was carried out in 50 µl 50 mM ammonium bicarbonate. First, we added 100 ng LysC (FUJIFILM Wako Pure Chemical Corporation) incubating 2 h at 37 °C followed by addition of 500 ng Trypsin (Promega) with over-night incubation at 37 °C. After digestion, peptides were sonicated for 1 min (water bath sonicator) and spun down. For peptide clean-up bead suspension was vacuum concentrated to 5 µl, adjusted to >90% acetonitrile

concentration and washed twice with neat acetonitrile, incubating 15 min at RT each time. Lastly, samples were eluted by addition of ultra HPLC-grade water (Pierce) in three steps, pooling the eluates. Eluted samples were concentrated in *vacuo* and resuspended in 0.1% formic acid prior to nanoLC-MS.

**Liquid chromatography-mass spectrometry**. Mass-spectrometry was done on a Q Exactive mass spectrometer coupled to an EASY-nLC 1000 liquid chromatography system (both Thermo Fisher Scientific) essentially as described in ref. [78] with modifications detailed below. Samples were injected twice. For the first injection, a nonlinear gradient was applied: 5 min: 5%, 40 min: 60%, 4 min: 80% (at a flow rate of 250 nl/min). A column wash out step followed this: 5 min: 80% B buffer (flow rate 500 nl/min). The gradient for the second injection was: 5 min: 10%, 40 min: 40%, 4 min: 80% (250 nl/min flow rate). This was followed by a wash out step: 5 min: 80% B buffer (flow rate 450 nl/min). Measurements were carried out in data-dependent mode employing the "sensitive method" as described previously[79].

**ATAC-seq**. ATAC-seq protocol was performed as described previously by ref. [80] with modifications in sample preparation as described in the embryo collection section. Fixed embryos were then thawed in 1 ml bufferA (60 mM KCl, 15 mM NaCl, 15 mM HEPES [pH 7.6], 4 mM MgCl2) and crushed with ultrasound (Covaris E220, 45 s, peak power 75, duty factor 10, cycles burst 200). Nuclei were pelleted by centrifugation ($3200 \times g$, 10 min at 4 °C).

We proceeded using the protocol of Buenrostro et al., 2015. Briefly, nuclei were resuspended in transposition reaction. Samples were then incubated at 37 °C for 30 min and immediately purified using the Qiagen MinElute PCR Purification Kit, according to manufacturer's instructions. Transposed DNA was eluted in 10 µl of elution buffer (buffer EB, Qiagen MinElute PCR Purification Kit) and amplified as stated in ref. [80]. All the sequences of custom primers were taken from ref. [81]. Next, the number of PCR cycles for each sample was determined using qPCR as recommended by ref. [80], and the remaining 45 µl PCR reaction was further amplified. DNA from these libraries was then purified using AMPure beads (Beckman Coulter, A63881) following the manufacturer's instructions. The quality of the purified libraries was assessed with Fragment Analyzer (Advanced Analytical).

**MNase-seq**. MNase-seq data presented in Fig. 1a, c was obtained from ref. [22]. MNase-seq data of Fig. 4a, b was produced for this manuscript, with two replicates per genotype. This protocol was modified from the Cut and Run protocol published by ref. [82]. Embryos were smashed using a pestle in 50 µl of the MNase wash buffer (20 mM HEPESpH7.5, 150 mM NaCl, 0.5 mM Spermidine, 1× Protease inhibitor cocktail), then rinsed with 150 µl of MNase wash buffer and centrifuged ($600 \times g$, 3 min at room temperature). Cells were wash afterwards in 500 µl MNase wash buffer and resuspended in 1 ml of MNase wash buffer. For eah sample, 10 µl of Concanavalin A (Polyscience, 86057-3) beads were added and incubated for 15 min under rotation. The beads were previously equilibrated with binding buffer (20 mM HEPES pH7.5, 10 mM KCl, 1 mM CaCl2, 1 mM MnCl2). Afterwards, the beads were resuspended in 1 ml permeabilization buffer (20 mM HEPES pH 7.5, 150 mM NaCl, 0.5 mM Spermidine, 0.05% Digitonin, 2 mM EDTA, 1× Protease inhibitor cocktail) and incubated for 2 h under rotation. Beads were washed twice with 1 ml digitonin wash Buffer (20 mM HEPES pH7.5, 150 mM NaCl, 0.5 mM Spermidine, 0.05% Digitonin, 1× Protease inhibitor cocktail). Next, 100 µl of 37 °C preheated digitonin wash buffer + MNase (20U, NEB M0247S) was added to each sample. Immediately after this, 3 µl of 100 mM CaCl2 were supplemented to each sample. Samples were then incubated at 37 °C for 30 min. To stop the reaction, 100 µl of stop buffer (340 mM NaCl, 20 mM EDTA, 4 mM EGTA, 0.05% digitonin, 50 µg/ml RNaseA, 25 µg/ml glycogen) was added. Samples were then incubated for another 30 min at 37 °C. Chromatin was purified by adding 0.1%SDS and 20 mg/ml of Proteinase K to each sample, followed by 1 h incubation at 50 °C. DNA was afterwards purified and large fragments (>500 bp) were removed using 0.5× volume of NucleoMag® NGS beads (Macherey-Nagel, 744970.50). The ChIP DNA Clean&Concentrator Kit (Zymo Research) was used to concentrate the samples according to manufacturer's instructions. Digestion efficiency was assessed by capillary electrophoresis on the Fragment Analyzer (Advanced Analytical) before preparing libraries. Libraries were prepared using the NEB Ultra II DNA Library Prep Kit for Illumina (E7645S and E6440) following the manufacturer's instructions. Integrity and size-distribution of the samples was assessed also after library preparation by running on Fragment Analyzer (Advanced Analytical).

**ChIP-seq, RELACS ChIP-seq, ATAC-seq and RNA-seq data processing**. Raw short read sequencing was processed uniformly using default parameters of *snakePipes-v2.0.2*[83]. RNA-seq was processed using the mRNA-seq workflow (–trim option). ChIP-seq, RELACS and ATAC-seq were processed with the DNA-mapping workflow (–trim, --dedup, –properPairs), followed by assay-specific workflow, i.e., ChIP-seq and RELACS with the ChIP-seq workflow and ATAC-seq by the ATAC-seq workflow. All libraries were mapped to dm6, using the Ensembl version 96 annotation[84].

**Blacklisting of hyper-accessible regions**. ATAC-seq was used to identify hyper-accessible regions. For that, we used peaks called using *snakePipes-v2.0.2*, ATAC-seq Genrich peak calling module (see "ChIP-seq, RELACS ChIP-seq, ATAC-seq and RNA-seq data processing" above). Peaks with a distance of up to 1 kb were merged and regions covered by merged peaks spanning more than 12 kb were discarded from downstream analysis.

**GRO-seq data processing and analysis**. In order to match the custom GRO-seq protocol, an in-house protocol was developed. The workflow processes GRO-seq single-end library, and retains UMI-like randomers that are part of the custom GRO-seq protocol. First, adapters were trimmed using *Cutadapt-v2.5*[85] (*-a "NNNN"<TGGAATTCTCGGGTGCCAAGG > --overlap=3 --minimum-length=12 --max-n 0*). UMI-like tetramers were trimmed and retained using *UMI-tools-v1.0*[86]. The SE library was mapped afterwards using *Bowtie2-v2.3.3.1*[87] with default parameters and bowtie2 *–local* alignment. Only alignments with MAPQ3 or higher were kept. For sequence read quality control *samtools-v1.10.0*[88], *deeptools-v3.3.1*[89], *FastQC-v0.11.5*[90] and *MultiQC-v1.8*[91] were used.

**Identification of active promoters from GRO-seq**. Active promoters from GRO-seq were identified using strand-specific *MACS2- v2.1.2*[92] peak calling (*--nomodel --extsize 100 -q 0.05 --call-summits*) on pooled Ctrl samples. For that, alignments were first split by strand (*sambamba -q view -F "not reverse_strand and sambamba -q view -F "reverse_strand"*, version 0.7.0); then, peaks were called and annotated according to the strand. Finally, the strand-specific peaks were pooled. A promoter was considered active if it contained a *MACS2* peak on the same strand and within 150nt distance. Active promoters were identified only for genes annotated as protein coding.

**GRO-seq differential gene expression analysis**. For differential gene expression analysis from GRO-seq a tailored *DESeq2-v1.26* workflow[93] was run. First, strand-specific read counting per gene using *featureCounts* from *subread-v1.5.3*[94] (GRO-seq; *–s 1 –Q 3*) on non-deduplicated alignments was performed, followed by *DESeq2* analysis. For the *DESeq2* analysis, genes with less than 10 reads on average across replicates and condition were removed and size factors were computed on our Zelda zygotic target genes set (see below).

**Maternal/Zygotic classification and Zelda zygotic target definition**. Zelda zygotic target genes were defined from Zelda targets of the Pol II occupancy analysis[10] (see "Zelda-dependent promoters identification and Differential Pol II occupancy analysis"), RNA-seq of Ctrl unfertilized eggs and Ctrl GRO-seq embryos. First, genes were classified as zygotic (GRO-seq only, Stage 5 expressed only), maternally deposited (RNA-seq, unfertilized eggs only) or maternal-zygotic, if present in both. For this, TPMs of RNA-seq from Ctrl unfertilized eggs were computed (transcripts per million[95]) (Supplementary Data 4) and genes with more than 5 TPMs in at least 3 replicates were retained. Genes were classified as Zelda-zygotic, if they had TPM > 5 on Ctrl embryos, were Zelda dependent, zygotically expressed and active at ZGA by GRO-seq.

**H2A.Z RELACS quantification**. The quantification H2A.Z was performed a) using the local scaling factors as defined in ref. [51], with the focus on active promoters (±400 nt around TSS) and b) using the *DEseq2* rlog transformation. For the local scaling factors, *deeptools-v3.4.1' multiBamSummary*[89] was use to count paired sequence reads in the defined promoters (*multiBamSummary BED-file –extendReads*) followed by the library-size corrected double ratios of IP and Input as described previously[51]. For the promoter quantification of H2Av RELACS IP and Input, reads per promoter were counted using *featureCounts*[94] (*-t promoter -g transcript_id -f -O -Q 3 --primary -s 0 -p -B, subread-v1.5.3*). Then, for the quantification, *DEseq2-v1.2.6* was used to compute the rlog transformation of the read counts. Since RELACS is quantitative, the size factors for library size correction were computed from the input only. Finally, rlog values were averaged between replicates.

**Cut&Tag/Run processing and quantification**. The quantitative Cut&Tag sequencing libraries contain DNA from *D. melanogaster* as well as a spike-in of *lambda* phage genome (Genbank: J02459.1). Here, the spike-ins were used to rule out a global change in chromatin abundance. The sequencing libraries were mapped to a hybrid dm6 and *Lambda* phage genome using *snakePipes-v2.4.3* DNA-mapping with updated *Bowtie2* mapping parameters (as in ref. [96]), as well as adapter trimming and MAPQ ≥ 3 filtering (*DNA-mapping –mapq 3 –trim –properPairs –dedup –alignerOpts='—local –very-sensitive-local –no-discordant –no-mixed -I 10 -X 700'*)[83]. Library size corrected for the Zelda IP signal tracks as produced by *snakePipes* were averaged and used for visualization in the heatmap shown on Fig. 2f.

After alignment and filtering, scaling factors from spike-ins were computed using *deeptools-v3.5.0*[89] multiBamSummary (*multiBamSummary bins –binSize 1000 –region <lambda phage genome coordinates> –scalingFactors*). The resulting scaling factors were used downstream to produce normalized signal tracks

(*bamCoverage -b <sample> --scaleFactor <sample scale factor>*) as well as in the quantification.

For the quantification, the number of PE reads were counted in 500 nucleotide bins across the dm6 genome (*multiBamSummary bins –binSize 500 –extendReads –outRawCounts <file>*). This was followed by the *DESeq2-v1.2.6* analysis using the previously computed spike-in scaling factors for size factor. Bins less than or equal to 25 read counts (average across samples) were discarded. The design matrix was setup to compare the samples by condition and correct for replicate effects (*design = ~replicate + condition*). In the last step, since there was no global change, we executed the *DESeq2* shrinkage of log2 fold changes (*type = 'normal'*).

**Identification and classification of H2A.Z positive promoters**. In order to identify promoters enriched for H2A.Z, *NucHunter*[97] was used to predict nucleosome positions from the H2A.Z ChIP-seq data on wild type samples. In order to do that, alignments were filtered for their quality using 'samtools view -q 3' from *samtools-v1.10.0*[88]. Fragment length of the filtered alignments were predicted using *fraglen* tool of *NucHunter*, these fragment lengths were used to run *callnucs* tool of *NucHunter* to predict the position of nucleosomes. Plus 1 nucleosomes (+1) were then predicted using the following criteria: 1) A nucleosome within 350 nucleotides from a TSS was predicted for at least one H2A.Z ChIP-seq replicate or 2) A nucleosome within 600 nucleotides from a TSS was predicted in both replicates, with less than 80 nucleotides difference from the center of the predicted nucleosomes between the two replicates. A promoter was considered H2A.Z positive, if there was a predicted +1 nucleosome overlapping or downstream the same promoter. The -1 nucleosomes were identified in the same way, but scanning upstream of the TSS.

**Zld-dependent promoters identification and differential Pol II occupancy analysis**. Differential Pol II occupancy was determined using the method described in ref. [10]. In brief, reads were countered per transcript using *featureCounts* (*-t transcript -g transcript_id -f -O -Q 3 --primary -s 0, subread-v1.5.3*). Reads were counted multiple times if they overlap several isoforms. Then, per transcript read counts were processed with *edgeR-v3.28.1*[98] to compute differential Pol II occupancy. Like in the published method, transcripts spanning less than 125 nucleotides were discarded and only transcripts with read counts across replicates sum more that 10 reads were considered. This method was used to identify Zelda targets in dm6 (logFC < −1 and FDR < 0.01) using published Pol II ChIP-seq data in Zld mutants[10]. Pol II occupancy per transcript was quantified using *edgeR*'s FPKM calculations. This same method was also used to quantify Pol II in DomKD and Ctrl ZGA embryos, with the following modifications: First, *edgeR norm.factors* were computed on our Zelda zygotic targets (see "Maternal/Zygotic classification and Zelda zygotic target definition" above). Differential transcripts were identified using the likelihood ratio test (edgeR::glmLRT). Then, unique promoters were identified and selected based on the most significant change among all transcripts for a given promoter (min FDR, edgeR results).

**Motif enrichment analysis**. Motif enrichment analysis was performed using *MEME-suite-v5.0.2* AME[99] with default parameters. Unique promoters were used. The core promoter analysis was performed on core promoters defined and constructed as -300 nt to 100 nt around the TSS. The core promoter motif database was constructed from Haberle & Stark, 2019[52] complemented with Ramalingam et al., 2021[100] as well as a known Zelda motif (Motif ID: MA1462.1)[101]. The TF motif analysis was performed using regions -600nt to 400nt around the TSS using Jaspar 2020 motifs[101]. For each of the three groups: H2A.Z positive, H2A.Z negative and Zelda-dependent, the corresponding background set was constructed from the other two groups.

**Gene ontology analysis**. Gene Ontology analysis was performed using *Metascape*[56]. The software of *Metascape* is freely available at: https://metascape.org and was used with default parameters. For Zelda-dependent and H2A.Z negative analysis, a single list containing all genes in the group was used for each. For H2A.Z positive genes, 8 random lists with 1000 genes each were generated and analyzed in the "Multiple gene list" mode. Prior to visualization, the observed gene count was translated into a gene ratio (number of observed / number of genes in a particular term). For visualization, the top 5 or the top 20 significant GO terms were selected per group and ordered using hierarchical clustering on the distance of -log10 *p*-value.

**Mass spectrometry data analysis**. Raw data was analyzed using *MaxQuant v1.6.14.0*[102]. Files were searched against a *Drosophila melanogaster* UniProt database containing Swiss-Prot and TrEMBL sequences (23,370 entries) plus an extended version of the *MaxQuant* contaminant database. Initial mass tolerance was 20 ppm followed by 4.5 ppm for main search and fragment tolerance of 25 ppm. Trypsin/P and D.P were used as enzymes and up to 2 missed cleavages were allowed. Carbamidomethylation of cysteine was used as fixed modification. Variable modifications included oxidation (M), deamidation (N, Q), acetylation of protein N-termini, acetylation (K), and phosphorylation (S, T). With only the first three variable modifications used for quantification, inter-sample relative abundance was determined using MaxLFQ[103] with enabling the match between runs

option (matching time window of 0.5 min). Intra sample abundance was approximated by iBAQ score[104] calculation. Peptide and protein FDR were kept at 1%. All other settings were kept at default. Downstream analysis was done in R-v4.0.3[105], using an in-house developed script employing the DEP package as base[106]. Briefly, contaminants, reverse, and only identified by site entries were filtered out. At least 2 valid quantitation values in any group (Bait or control) were required. Data was vsn-normalized and missing values were imputed by drawing values from a defined distribution (width 0.5, downshift 1.8) as described previously[107]. Statistical analysis was done using limma-v3.44.3 (with trend = TRUE)[108], and the obtained p-values were corrected for multiple hypotheses by Benjamini–Yekutieli[109]. Differentially enriched proteins were classified by having an adjusted p-value ≤ 0.05 and a fold change ±50% bait/control.

**MNase-seq data processing and analysis**. MNase-seq data presented in Fig. 1a and Fig. 1c was obtained from ref. [22]. From this published data set, MNase-seq data from two different developmental stages: stage 4 (pre-ZGA) and stage 5 (ZGA) were processed and analyzed. Briefly, the quality of the data has been checked and the data was mapped using the DNA-mapping pipeline of snakePipes-v2.1.0[83] with the following parameters: --trim --properPairs --dedup --mapq 3 --fastqc. For the purpose of visualization, three replicates of samples from each of the developmental stages have been merged after assessing their correlations. The above pipeline has been run with the same parameters on the merged samples. The coverage files were then computed for each stage using bamCoverage from deeptools-v3.4.3[89] with the following parameters: --binSize 1 -p 50 --effectiveGenomeSize 142573017 --normalizeUsing RPGC --MNase along with --ignoreForNormalization to remove the blacklisted regions and to minimize the bias in calculating the genome-wide coverage. Blacklisted regions can be found at https://github.com/iovinolab/dom-study_2020. MNase-seq data of Fig. 4a and Supplementary Fig. 4a was produced for this manuscript. This data contains both Ctrl and DomKD samples, two replicates per condition and has been mapped and pre-processed using the same steps as above. However, to efficiently capture +1 nucleosomes, filtered reads obtained from running the DNA-mapping pipeline on merged samples, were further filtered for read length. Only those with a length between 130 and 200 base pairs were retained and used to compute the coverage files. The alignmentSieve module of deeptools-v 3.4.3[89] has been used for this filtering step.

**HiC data processing and TAD insulation score calculation**. HiC data from both wildtype and Domino knockdown, two replicates per condition, have been mapped on dm6 genome and processed using the HiC pipeline of snakepipes-v2.1.0. Matrices of 2 kb resolution (--binSize 2000) have been created, replicates were merged after assessing their correlation (QC of the pipeline, distance vs. counts and visualizing matrices). Merged matrices were balanced using KR method (--correctionMethod KR). To call TADs hicFindTADs form hicexplorer-v3.4.3[110] was used with the following parameters: --thresholdComparisons 0.01 --correctForMultipleTesting bonferroni --minDepth 20000 --maxDepth 100000 --step 2000 --delta 0.01. TAD calling generates a bedgraph file that contains the insulation scores for each condition. These files have been converted to bigwig files using bedGraphToBigWig-v4 as one of the UCSC utility tools[111]. Fastq files of published HiC data on both control and Zelda knockdown from Hug et al., 2017[23] were downloaded and processed identically to the in-house data. Afterwards, bigwigCompare with --operation subtract --skipNonCoveredRegions from deeptools-v3.4.3[89] has been used to generate a difference coverage file and presents the subtraction of the insulation score of control data from Zelda knockdown.

**Data analysis software**. All remaining data data transformation, analysis and visualization have been execute using R-v3.6.3[105], ggplot-v3.28.1[112]. For operations of genomic regions, bedtools-v2.27.0[113] was used.

**Reporting summary**. Further information on research design is available in the Nature Research Reporting Summary linked to this article.

## Data availability
The data that support this study are available from the corresponding author upon reasonable request. The sequencing data generated in this study have been deposited in the NCBI's Gene Expression Omnibus and are accessible through GEO Series accession number GSE173240. The mass-spectrometry proteomics data generated in this study have been deposited to the ProteomeXchange Consortium via the PRIDE partner repository with the dataset identifier PXD029061. MNase-seq data presented in Fig. 1a, was obtained from ref. [22], and is accessible through GEO accession numbers: GSM3736319, GSM3736320, GSM3736321, GSM3736322, GSM3736323, GSM3736324, GSM3736325. RNA-seq data presented in Supplementary Fig. 3m–o was obtained from ref. [57] and is accessible through SRA: SRP001696. Specific samples used for this study have the accession numbers SRR1197327 to SRR1197338 and SRR1197363 to SRR1197370. HiC data presented in Supplementary Fig. 4c, d was obtained from ref. [23]. and is accessible through ArrayExpress: E-MTAB-4918. Specific samples used for this study have the accession numbers ERR1533226 to ERR1533236, and ERR1912884 to ERR1912887. Enhancer candidates used in Supplementary Fig. 1b were downloaded

from https://enhancers.starklab.org/ based on ref. [41]. Source data are provided with this paper, which includes raw data for graphs and gel blot images for Figs. 2d, 3c, d, Supplementary Figs. 2d–f, 3d–h, 3k, 3n and 3o in the Source Data file. Promoter classification for Fig. 1 and Supplementary Fig. 1, and analysis files for Figs. 2b, 3a, b, Supplementary Fig. 3c, i, j are provided as Supplementary Data. All other data are available within the article and its Supplementary Information. Source data are provided with this paper.

## Code availability
Custom code for sequencing data analysis is accessible through the GitHub repository: https://github.com/iovinolab/dom-study_2020.

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

## Acknowledgements

We would like to thank our colleagues Peter Becker, Alessandro Scacchetti and Kenneth Börner from LMU, Munich for kindly sharing the DomA-, DomB-, and deltaDom-GFP-Flag flies, for useful feedback and for critical reading of the manuscript. We also acknowledge Carla Margulies her generous gift of Rpb3 antibody, Melissa Harrison her generous gift of Zelda antibody, Vanja Haberle and Alexander Stark for providing the core promoter motif database. We are particularly grateful to Eva Loeser, Melanie Schaechtle, Nazerke Antinbayeva and Filippo Ciabrelli from the Iovino lab, the Bioinformatics and Sequencing facilities at the Max Planck Institute of Immunobiology and Epigenetics (MPI-IE); as well as the Imaging facility, Proteomics facility (Gerhard Mittler, Witold Szymanski and Yaarub Musa), and Fly facility at MPI-IE. The Bloomington Drosophila Stock Center (NIH P40OD018537) and the Transgenic RNAi Project at Harvard Medical School (NIH/NIGMS R01-GM084947) provided fly stocks used in this study. Shelby Blythe for useful feedback and sharing fly stocks. We would also like to thank all members of the Iovino lab for critical feedback on the manuscript. D.I.-M., F.Z., M.S.-S. and F.C. are supported by the Max Planck Society and IMPRS program. N.I. is supported from the Max Planck Society; DFG:CRC992, Project B06; Behrens-Weise Stiftung; CIBSS - EXC 2189; Deutsche Forschungsgemeinschaft - Project ID 192904750 - CRC 992 Medical Epigenetics. Also, this project has received funding from the European Research Council (ERC) under the European Union's Horizon 2020 research and innovation programme (grant agreement No.819941) ERC CoG, EpiRIME.

## Author contributions

D.I. performed the ChIP-seq, ATAC-seq, RELACS ChIP-seq, RNA-seq experiments, phenotypical characterization, sample collection, immunostainings, microscopy data collection and initial computational analysis. M.R. performed all the computational analysis related to ChIP-seq, ATAC-seq, GRO-seq, RELACS and Cut&Tag data. P.Q. performed the GRO-seq experiments and contributed to the computational analysis. L.R. performed all the HiC and MNase-seq computational analysis. F.Z. performed the HiC experiments. M.S. contributed with the IP-MS experiments. F.C. performed the Cut&Tag experiments. A.G. contributed with proteomics data analysis. G.C. contributed with the design and analysis of GRO-seq experiments. N.I. design and supervised the project with inputs from D.I. N.I. and D.I. wrote the manuscript.

## Funding

## Competing interests

The authors declare no competing interests.
