## [Peer Review File · Nature Communications]

REVIEWERS' COMMENTS

Reviewer #1 (Remarks to the Author):

The manuscript is much approved and publication is supported. Here are a few suggestions:

Prior major concern:

2. The authors show add more evidence that Domino regulates H2A.Z association with the chromatin. How does H2A.Z localization change in the DomKD background; in particular, during mitosis does H2A.Z stay associated with the sister chromatids or does it diffuse away?

Line 185: "Yes, we did not observe any change in the levels of phospho-H2A.Z". Hard to see anything in ext data Fig 2i... The change in p-H2A.Z in wt versus DomKD should be quantified and number of embryos examined, noted.

Are H2A.Z levels also decreased in the DomKD embryos at nc14? (i.e. ext data fig 2h,i). Only the change at early nuclear cycles is shown. Perhaps remove the p-H2A.Z data and add nc14 for H2A.Z in DomKD?

Add'l suggestions:

Several studies have identified other pioneer-like factors such as Odd paired (Opa) that act with Zelda to manage the MZT (Koromila et al., eLife 2020; Soluri et al., eLife 2020). While it would be beyond the scope of the current study to compare H2A.Z to Opa, it is possible that H2A.Z relates to function of this other pioneer factor whose binding was also shown to correlate with H3K4me3 (see Koromila et al.) and should be mentioned.

Line 371: Are Opa binding sites enriched in H2A.Z positive genes?

Line 349: "precedes Pol II binding to chromatin, suggesting that it primes genes for transcriptional activation". Add figure call here to support statement. Fig. 1C?

Line 350: provide context for the statement. How does H4K12 not being affected suggest a role for H2A.Z at promoters? Add reference?

Line 395: the point of this sentence was unclear "not all transcriptionally activated zygotic genes are affected by changes in core histone levels"... what is connection to H2A.Z?

Reviewer #2 (Remarks to the Author):

In the revised manuscript Ibarra-Morales add several important additional experiments and analysis that significantly strengthen the manuscript including analysis of H2A.Z in enhancers, analysis of Zelda occupancy in domino KD embryos, phospho-H2A.X staining, comparisons with previous lists of ZGA genes, and a comparison of the H2A.Z positive vs H2A.Z negative promoters.

With these additional experiments and analysis the story has come into somewhat better focus. H2A.Z and Zelda appear to be parallel and independent pathways with H2A.Z functioning primarily or solely at promoters. The evidence is more circumstantial for GAGA, but paints a somewhat similar picture of independence.

The new comparison between H2A.Z positive and H2A.Z negative promoters indicates that H2A.Z incorporation does not predict the timing of future expression Fig 3o except for re-expression of maternally provided genes well after where one might conventionally call "ZGA". This result substantially undermines the initial "master regulator" language used in the initial submission. However, the authors have already modulated that claim in the current draft. It might be interesting to subtract the zelda (and GAGA) dependent genes from the H2A.Z negative gene list to see if there is some type of "OR" function involved in this process- eg. genes need H2A.Z OR Zelda (OR GAGA-factor) to activate. Further probing in this direction may yield a more complete picture of how these factors contribute to ZGA.

The comparison with previous ZGA datasets is helpful. The limited agreement between the gro-seq and RNA-seq defined ZGA gene lists is frustrating. However, that is by no means a weakness of this paper (it is a general problem that has persisted in the field that we cannot agree on exactly which genes are the true ZGA genes).

Overall, I continue to believe that these data will be highly useful for future comparison and now feel that the author's interpretations are well supported by their data.

Point by point response:

Histone variant H2A.Z regulates zygotic genome activation

Summary

We would like to thank all the reviewers, again, for their constructive comments throughout the revision process. In this revised version of the manuscript, we have now responded to all the final reviewer's suggestions and we believe that the manuscript is considerably improved.

Reviewers' Comments:

Reviewer #1 (Remarks to the Author):

The manuscript is much approved and publication is supported. Here are a few suggestions:

Prior major concern:

2. The authors show add more evidence that Domino regulates H2A.Z association with the chromatin. How does H2A.Z localization change in the DomKD background; in particular, during mitosis does H2A.Z stay associated with the sister chromatids or does it diffuse away?

Line 185: "Yes, we did not observe any change in the levels of phospho-H2A.Z". Hard to see anything in ext data Fig 2i... The change in p-H2A.Z in wt versus DomKD should be quantified and number of embryos examined, noted. Are H2A.Z levels also decreased in the DomKD embryos at nc14? (i.e. ext data fig 2h,i). Only the change at early nuclear cycles is shown. Perhaps remove the p-H2A.Z data and add nc14 for H2A.Z in DomKD?

We have evidence from quantitative ChIP-seq (RELACS) and western blot that H2A.Z in DomKD is reduced on chromatin at nc14 (this is shown in Figure 2e, f and Supplementary Figure 2f). The quantitative measurement about H2A.Z levels on the chromatin at NC14 in control and DomKD are further validated by immunohistochemistry experiments. Supplementary Figure 2g shows a NC14 embryo in both Ctrl and DomKD embryos, stained for H2A.Z. All this evidence supports the fact that H2A.Z is reduced at NC14.

Regarding phospho-H2A.Z, we have repeated the immunostainings several times for independent samples and always observed similar results. We do not have, however, any image quantification. As reviewer #2 previously requested to mention phospho-H2A.Z, we decided to keep the image in Sup. Fig.2i.

Add'l suggestions:

Several studies have identified other pioneer-like factors such as Odd paired (Opa) that act with Zelda to manage the MZT (Koromila et al., eLife 2020; Soluri et al., eLife 2020). While it would be beyond the scope of the current study to compare H2A.Z to Opa, it is possible that H2A.Z relates to function of this other pioneer factor whose binding was also shown to correlate with H3K4me3 (see Koromila et al.) and should be mentioned.

We appreciate the reviewer's insight on this topic. We added it now in the discussion, lines 418-419.

Line 371: Are Opa binding sites enriched in H2A.Z positive genes?

We checked for Opa binding using the JASPAR database reference motif and we found no enrichment of Opa consensus binding sites in any of the three groups (H2A.Z positive, H2A.Z negative or Zelda-dependent).

Line 349: “precedes Pol II binding to chromatin, suggesting that it primes genes for transcriptional activation”. Add figure call here to support statement. Fig. 1C?

Thanks for pointing this out. We added the figure call in line 362

Line 350: provide context for the statement. How does H4K12 not being affected suggest a role for H2A.Z at promoters? Add reference?

Depletion of both Domino isoforms affect only H2A.Z deposition on chromatin and not H4K12ac (as shown in Supplementary Figure 2f, j). Therefore, we believe this data suggests that the phenotype we observe is mainly due to the lack of H2A.Z, since the other activation mark, H4K12ac, previously associated with Domino, is not affected.

Moreover, we do not find any evidence of H2A.Z enrichment at enhancer regions as shown in Supplementary Figure 1b. Therefore we suggest that H2A.Z function mainly at promoter regions (see Sup. Fig. 1a). The reference in line 363 points to the paper where a reduction in H4K12ac is observed upon DominoA knockdown.

Line 395: the point of this sentence was unclear “not all transcriptionally activated zygotic genes are affected by changes in core histone levels”... what is connection to H2A.Z?

We thank the reviewer for pointing this out and we apologize for the lack of clarity. In this part of the discussion, we sought to discuss previous work in which was found that affecting overall histone levels had an effect over ZGA. We agree that there is no a direct connection with H2A.Z and decided to eliminate this paragraph from the discussion.

Reviewer #2 (Remarks to the Author):

In the revised manuscript Ibarra-Morales add several important additional experiments and analysis that significantly strengthen the manuscript including analysis of H2A.Z in enhancers, analysis of Zelda occupancy in domino KD embryos, phspoho-H2A.X staining, comparisons with previous lists of ZGA genes, and a comparison of the H2A.Z positive vs H2A.Z negative promoters. With these additional experiments and analysis the story has come into somewhat better focus. H2A.Z and Zelda appear to be parallel and independent pathways with H2A.Z functioning primarily or solely at promoters. The evidence is more circumstantial for GAGA, but paints a somewhat similar picture of independence.

The new comparison between H2A.Z positive and H2A.Z negative promoters indicates that H2A.Z incorporation does not predict the timing of future expression Fig 3o except for re-expression of maternally provided genes well after where one might conventionally call “ZGA”. This result substantially undermines the initial “master regulator” language used in the initial submission. However, the authors have already modulated that claim in the current draft. We thank the reviewer for his/her insightful critic of the manuscript, which resulted in a much improved version.

It might be interesting to subtract the zelda (and GAGA) dependent genes from the H2A.Z negative gene list to see if there is some type of “OR” function involved in this process- eg. genes need H2A.Z OR Zelda (OR GAGA-factor) to activate. Further probing in this direction may yield a more complete picture of how these factors contribute to ZGA.

Supplementary Figure 3o does not contain Zelda-dependent genes. We now generated a new figure for the reviewer (below) that contains Zelda-dependent genes. In agreement with previous literature, we see the Zelda-dependent genes have in general a high expression, with a peak around ZGA (non-maternal 2-4hrs).

The comparison with previous ZGA datasets is helpful. The limited agreement between the gro-seq and RNA-seq defined ZGA gene lists is frustrating. However, that is by no means a weakness of this paper (it is a general problem that has persisted in the field that we cannot agree on exactly which genes are the true ZGA genes).

We agree that the overlap between the GRO-seq presented in this study is little with the RNA-seq presented in other studies before. These are different techniques and both of them have disadvantages. However, we believe that by using GRO-seq, we have a more comprehensive picture of the ZGA transcripts, especially with regard to lowly expressed genes that are generally not picked by RNAseq and that instead show a clear signal with our GRO-seq. We hypothesize that is the reason we obtain a higher number of maternal/zygotic transcripts compared to previous studies.

Overall, I continue to believe that these data will be highly useful for future comparison and now feel that the author's interpretations are well supported by their data.